# Generation of human colon organoids from healthy and inflammatory bowel disease mucosa

**Isabella Dotti** [ID]**, Aida Mayorgas, Azucena Salas** [ID]*

Department of Gastroenterology, IDIBAPS, Hospital Clínic, CIBER-EHD, Barcelona, Spain

* asalas1@recerca.clinic.cat

## Abstract

Ulcerative colitis and Crohn's disease are chronic inflammatory bowel diseases (IBD) of unknown cause characterized by a relapsing-remitting behavior. Growing evidence supports the idea that the epithelial barrier plays a central role in the pathogenesis of IBD as well as in its evolution over time, thus representing a potential target for novel therapeutic options. In the last decade, the introduction of 3D epithelial cultures from *ex vivo*-expanded intestinal adult stem cells (ASCs) has impacted our ability to study the function of the epithelium in several gastrointestinal disorders, including IBD. Here, we describe in detail a reproducible protocol to generate Matrigel-embedded epithelial organoids from ASCs of non-IBD and IBD donors using small colonic biopsies, including steps for its optimization. A slightly modified version of this protocol is also provided in case surgical samples are used. With this method, epithelial organoids can be expanded over several passages, thereby generating a large quantity of viable cells that can be used in multiple downstream analyses including genetic, transcriptional, proteomic and/or functional studies. In addition, 3D cultures generated using our protocol are suitable for the establishment of 2D cultures, which can model relevant cell-to-cell interactions that occur in IBD mucosa.

## Introduction

Crohn's disease (CD) and ulcerative colitis (UC) are chronic inflammatory bowel diseases (IBDs) that affect the gastrointestinal tract with alternating periods of activity and remission [1]. Among the multiple factors associated with IBD development, there is increasing evidence supporting the essential role of the intestinal epithelial barrier in the pathogenesis, prognosis, and perpetuation of these inflammatory disorders [2, 3]. A methodology for generating long-term human 3D cultures from adult intestinal stem cells (ASCs) of gut epithelium was first established about a decade ago [4, 5]. These *ex vivo* cultures, also known as organoids, represent a promising tool for modeling gastrointestinal homeostasis and disease, including IBD [6–9], since they closely recapitulate the genetic and phenotypic characteristics of the epithelium from which they have been generated [10, 11].

Here, we describe a straightforward method for the generation and long-term maintenance of human epithelial organoids using colonic biopsies from both non-IBD and IBD donors. We

**Data Availability Statement:** All relevant data are within the article and its Supporting Information files.

**Funding:** This work was supported by the 3TR project, which received funding from the Innovative

Medicines Initiative 2 Joint Undertaking (https://www.imi.europa.eu/) as part of grant agreement No. 831434, and by an ECCO Grant (https://www.ecco-ibd.eu/), in 2019. The funders had and will not have a role in study design, data collection and analysis, decision to publish, or preparation of the manuscript.

**Competing interests:** The authors have declared that no competing interests exist.

also provide useful tips for its execution and the results of its systematic use in large cohorts of human biopsies from non-IBD and IBD patients. A slight adaptation to the protocol has been also included, for those instances in which a surgical sample is used as the starting material rather than biopsies. This approach was initially used for generating organoids from human healthy colonic mucosa of surgical samples [5]. It was then optimized in our laboratory, expanding its application to biopsy samples from IBD patients [12]. This protocol, unlike other currently available methods [13–15], relies on the use of dispase for culture passaging. While this strategy makes the execution of the passage step more time-consuming, in our hands it proved more successful than other organoid-dissociation methods (e.g., mechanical disruption, TrypLE). In fact, this protocol proved useful for expanding organoids from non-IBD and IBD donors for at least 6 passages with no evident decrease in growth efficiency. On average, the minimum time required for finalizing this protocol is 13 days, from sample collection to the complete differentiation of the first-passage crypts (Fig 1). If desired, the cultures can be cryopreserved for future use.

The resulting organoid cultures can be used to answer multiple clinically relevant questions. They can help identify the presence of lasting alterations in the epithelium of IBD patients [12, 16]. In addition, they can be used to explore the impact of IBD-specific epithelial mutations [17, 18], as well as the effects of environmental signals, on the epithelial response in IBD mucosa [19–23]. More recently, the establishment of monolayers from 3D organoids has expanded the potential use of primary epithelial cultures as a tool for exploring the complex mechanisms underlying the interaction of the epithelium with the luminal environment, including modelling bacterial infections [24–26]. Moreover, seeding dissociated organoids onto a Transwell insert, or even on a sophisticated organ-on-a-chip device, have facilitated the setting of co-cultures with other cell populations of interest [27, 28]. Finally, these organoid cultures may be used in a clinical setting as a model for testing novel therapeutic agents for IBD treatment [29], as has been successfully realized in other gastrointestinal disorders [30, 31].

Despite the versatility of this protocol, results obtained using organoid cultures are prone to experimental biases that should be taken into account. The major factors that can affect the

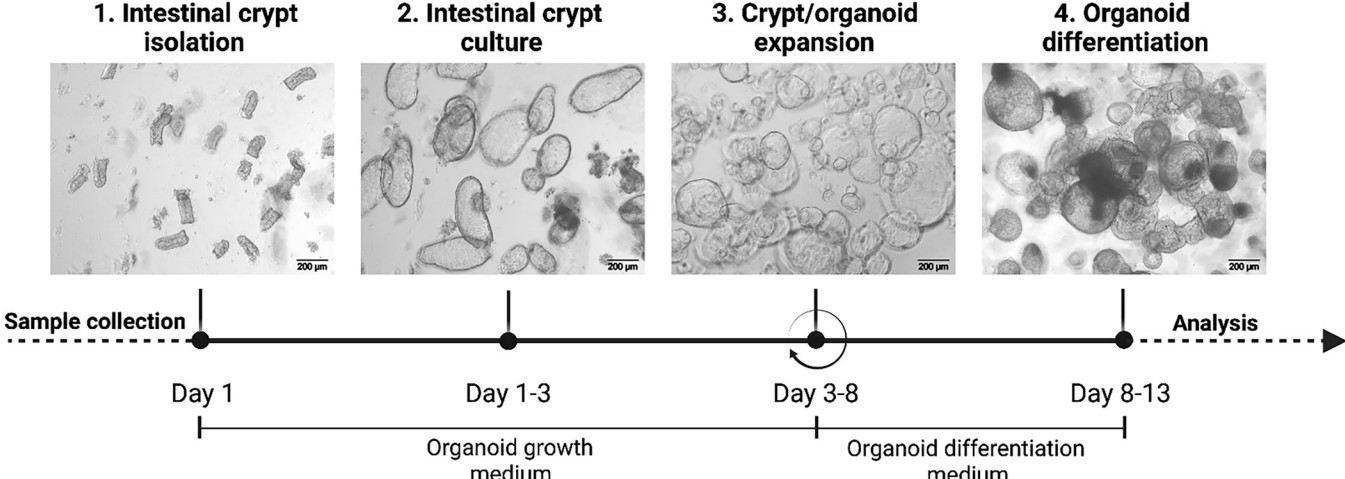

**Fig 1. Schematic representation of the experimental workflow from sample collection to 3D organoid generation and differentiation.** Running the first round of expansion (i.e., from crypts to organoids and their differentiation) takes 13 days on average. Step 3 can be repeated several times (↻) depending on the experimental plan. Once generated, both stem cell-enriched and differentiated cultures can be used for several downstream applications, including transcriptional analysis of any kind (from qPCR to single cell RNAseq), imaging, and protein analysis (e.g., Western Blot or ELISA assay). This figure was created using www.biorender.com.

reproducibility of the results include the intrinsic donor-to-donor variations and the batch-to-batch instability of the culture reagents (e.g., WNT3a-conditioned medium, Matrigel, recombinant proteins). Moreover, the organoid-to-organoid variability within a culture may also impact the results [32]. We thus recommend including at least 5–6 donors when *in vitro* stimulation experiments are planned, and at least 12–15 donors per group when comparative analyses between cohorts of patients (i.e., non-IBD vs IBD subjects) are planned. As organoids derived from different segments of the intestinal tract show intrinsic differences [11, 33], we also recommend using biopsies from the same anatomical region to minimize this variability. Of note, recent studies show that prolonged local inflammation promotes the accumulation of somatic alterations in the IBD epithelium [17, 34]. Thus, in clinical IBD cases with an extensive history of inflammation, comparisons between affected and unaffected areas, even from the same patient, could also be valuable for exploring the impact of somatic changes on the phenotype of IBD epithelium.

## Materials and methods

The protocol described in this peer-reviewed article is published on protocols.io (DOI: dx.doi.org/10.17504/protocols.io.rm7vz3wr4gx1/v2 -Private link for reviewers: https://www.protocols.io/private/F13BA2CE694A8C5FFCF0EBD83DC6FBE1 to be removed before publication-) and is included for printing as (S1 File) with this article.

The success rate of generating organoids through this lab protocol was determined using a database of 189 colonic biopsy samples collected between January 2016 and June 2021. These samples came from a cohort of pediatric (n = 45) and adult (n = 137) individuals, for a total of 182 subjects. Both non-IBD and IBD subjects were included. Pediatric subjects ranged between 8 and 16 years old, and adult subjects between 18 and 76 years old. Non-IBD controls were those subjects undergoing colonoscopy for mild gastrointestinal symptoms or screening for colorectal cancer, who had a normal mucosa and no history of IBD. Patients diagnosed with IBD (UC and CD) showed no evidence of IBD-associated dysplasia or neoplasia, as assessed by endoscopy. Endoscopic activity at the time of examination was categorized according to the endoscopic Mayo subscore for UC [35] and the Simple Endoscopic Score for Crohn's Disease (SES-CD) for CD [36]. Samples with endoscopic Mayo $\leq$ 1 and SES-CD $\leq$ 6 were used. The present protocol also describes an adaptation for generating organoids from surgical samples. In this case, samples were collected from the healthy colonic mucosa of adult non-IBD subjects that underwent surgical intervention for colorectal cancer. The healthy mucosa was collected at least 10 cm from the margin of the affected area.

Subjects were recruited at the Hospital Clinic, Barcelona, Hospital Mutua de Terrassa, Barcelona, and the pediatric Hospital de Sant Joan de Deu, Barcelona, Spain.

Approval from the ethics committee (Comité de Ética de la Investigación con medicamentos, Hospital Clínic, Barcelona-CEIm-) and written consent (specific for pediatric and adult donors) were obtained prior to human sample processing. In case of underage donors, the consent was signed by the legal representatives. All patients' personal information was anonymized. The subjects were not previously registered as organ donors. Medical costs were all covered by the Spanish public health system; no cash payments were provided to the family of the donor.

## Results and discussion

Here, we describe a robust protocol for generating epithelial 3D organoid cultures from human healthy and IBD colonic mucosa. This approach relies on the selective *ex vivo* expansion of epithelial adult stem cells (ASCs), located at the bottom of intestinal crypts (S1 Fig)

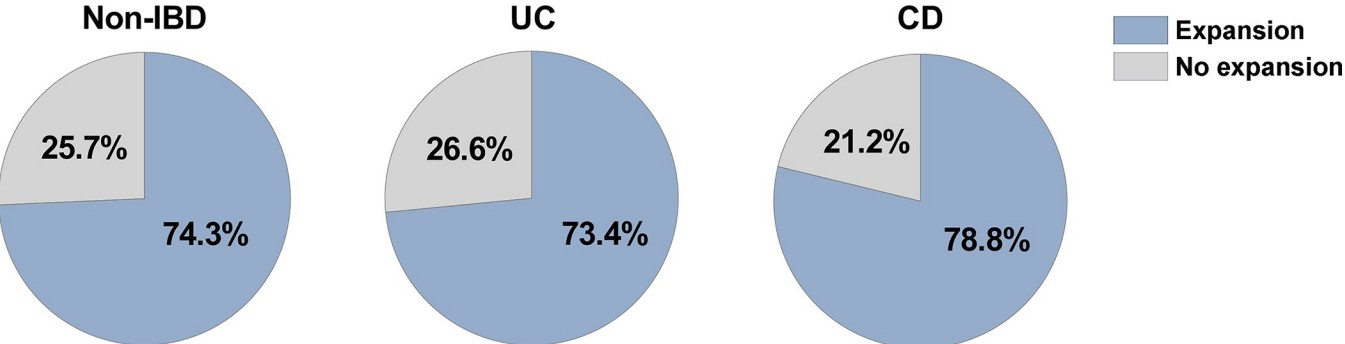

**Fig 2. Pie chart indicating the success rate of generating organoids from non-IBD and IBD (UC and CD) mucosa using the described protocol.** Data were retrieved from a cohort of biopsy samples collected over a 5-year period (2016–2021) and which were processed according to the described protocol. Biopsy samples in non-IBD, UC and CD groups are from pediatric and adult subjects (representing the 23.2% and the 76.8% of the total number of biopsies, respectively), and were obtained from different parts of the colon (i.e., ascending, descending and sigmoid colon). The percentages refer to the successful generation ("Expansion") of viable organoids after one passage (from crypts culture to organoids), with success being defined as the generation of a growing culture of 3D organoids after 5 days of expansion. In our hands, failure ("No expansion") primarily stemmed from crypt culture contamination due to microorganisms or from undetermined causes. For UC and CD patient cohorts, samples were derived from non-inflamed/mildly inflamed involved colonic mucosa. Total number of biopsy samples processed: non-IBD samples, n = 74; UC samples, n = 49; CD samples, n = 66.

[37], resulting from the use of specific medium conditions that recapitulate the crypt niche environment *in vivo*.

The method presented here is suitable for establishing organoid cultures using as starting material 4–8 biopsies from the colonic mucosa of non-IBD and IBD donors. With this method, comparable rates of success in generating organoid cultures, whether from non-IBD or UC/CD colonic samples, are possible provided that the biopsies are obtained from a mildly inflamed or a non-inflamed (i.e., uninvolved or previously affected) colonic region (see below for more details) (Fig 2).

The organoid culture, regardless of whether it is generated from a non-IBD or IBD donor, can be maintained using this protocol for at least 6 passages with no apparent decrease in the expansion rate. In our experience, failure to establish and/or expand 3D organoids using this protocol mainly occurs due to unpredictable factors such as the microbial contamination of the crypt culture (more frequent in IBD than non-IBD samples), undetermined donor-dependent issues (regardless of the inflammatory status of the sample), or the excessively small size of the biopsies.

The first step of the protocol is the mechanical isolation of the epithelial crypts from the biopsy samples. The release of single crypt units is achieved by EDTA-mediated chelation (Fig 1, Step 1 "Intestinal crypt isolation"). The isolation of healthy crypts may be affected by the sample characteristics. Indeed, in biopsy samples from the healthy mucosa of non-IBD donors, as well as in samples from the non-inflamed (or mildly inflamed) mucosa of IBD patients, there is no or limited inflammatory infiltrate and the epithelial layer is well preserved and organized (Fig 3), thus making viable crypts more easily isolated.

Conversely, in highly inflamed IBD intestinal samples epithelial erosion is often extensive, and a well-defined epithelial layer may not be distinguishable (Fig 3). This has a negative impact on the isolation of viable stem cell-containing crypts and, consequently, on the generation of successful organoid cultures. For this reason we recommend using UC biopsy samples with an endoscopic Mayo score $\leq$ 1 and CD biopsy samples from intestinal segments with a SES-CD $\leq$ 6. We have also observed that, overall, inflamed CD samples are more likely to give rise to healthy organoid cultures, compared to UC. We suppose that this may be due to the erosive nature of UC compared to CD, where the epithelial barrier is generally better

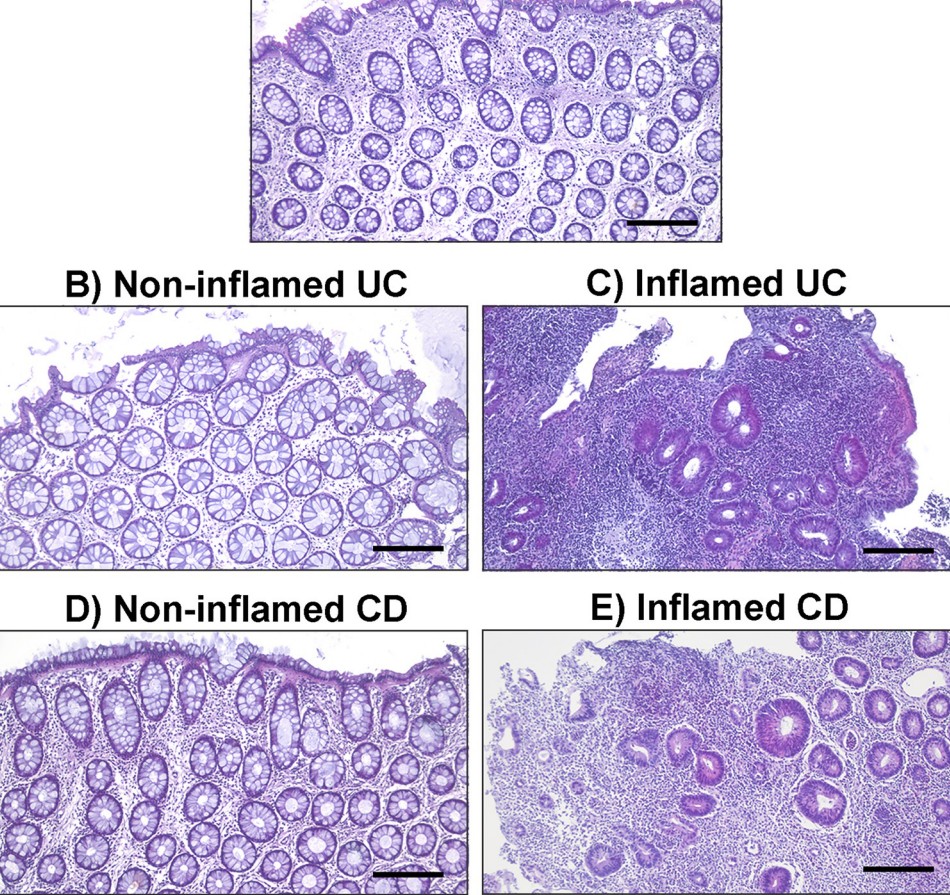

**Fig 3. Representative histological sections stained with hematoxylin and eosin of colonic biopsies from non-IBD and IBD donors.** The images show the marked histological alterations of the epithelial layer due to inflammation in active IBD mucosa, compared to the healthy and non-inflamed IBD (i.e., uninvolved or previously affected) mucosa. (A) Healthy sigmoid mucosa from a non-IBD subject; (B) Non-inflamed ascending colonic mucosa from a UC patient; (C) Inflamed sigmoid colonic mucosa from a UC patient; (D) Non-inflamed sigmoid colonic mucosa from a CD patient; (E) Inflamed ascending colonic mucosa from a CD patient. All samples are from adult donors. Scale bar: 200 μm.

preserved. Epithelial erosion can be even more profound in surgical samples due to the extensive ulceration of the mucosal surface characteristic of the severe and refractory disease in patients undergoing colectomy. For this reason, the surgical sample protocol included in this manuscript is mainly indicated for generating organoids from non-IBD donors (patients undergoing intestinal resection for non-inflammatory conditions).

Once isolated, crypts from both non-IBD and IBD donors are embedded in Matrigel and cultured in organoid growth medium (Fig 1, Step 2 "Intestinal crypt culture"), which contains a cocktail of factors capable of reproducing the crypt niche *in vivo*. Under these conditions, the bottom and mid-crypt start to swell within the following 1–3 days, while the most apical and differentiated part is lost (Fig 4A).

Swelling of the crypt lumen in culture, even if the crypt is broken during isolation (Fig 4B), is a good early predictor of the ASC niche's viability, and of the successful establishment of the organoid culture. After 2–3 days, the swollen crypts should be passaged. Indeed, leaving

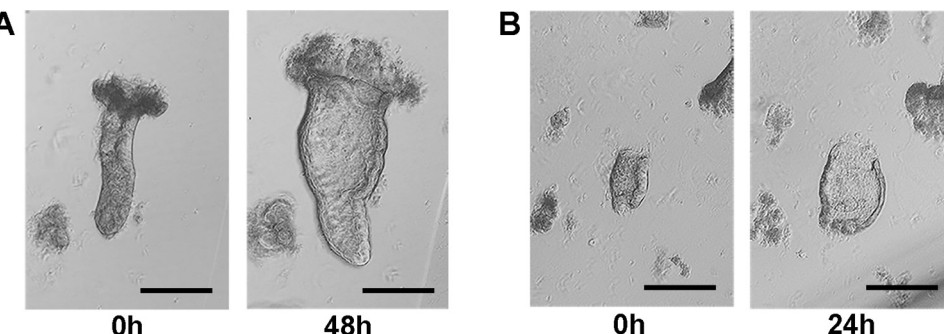

**Fig 4. Swelling of colonic crypts embedded in Matrigel and cultured in organoid growth medium for 24–48 hours.**
If the adult stem cell niche is preserved, swelling of the lumen will be observed in most viable crypts. (A) Swelling of an intact crypt derived from a non-inflamed sigmoid mucosa of a pediatric CD patient; (B) Swelling of a broken crypt derived from a non-inflamed sigmoid mucosa of an adult CD patient. Scale bar: 200 μm.

Matrigel-embedded crypts in culture longer substantially decreases their viability, as indicated by the accumulation of dead cells in the lumen of swollen crypts (S2 Fig). Of note, crypts embedded in Matrigel can also be used for short-term experiments not requiring further ASC expansion [38, 39].

Organoid expansion from a crypt culture (first passage) or from previously passaged organoids (Fig 1, Step 3 "Crypt/organoid expansion") should be performed taking into account the size, density and estimated viability rate of the culture to be expanded, regardless of whether or not the organoids are generated from non-IBD or IBD donors. A 1:3 dilution is usually suitable for expanding swollen crypts, while 1:4 and up to 1:7 can be used for previously passaged organoids. The re-seeding dilution of dissociated organoids is critical to the success of the expansion (Fig 5).

Indeed, if over-diluted, a single-cell culture does not provide the paracrine factors required for proper organoid growth. In contrast, if the cell density is excessively high, factor competition can lead to exhaustion of the culture. Under proper re-seeding dilution, signals from medium components induce the progressive expansion of the ASC population in the form of organoids without requiring a mesenchymal niche. After 5–6 days, the organoid culture usually appears as a heterogeneous population of cystic structures characterized by a thin and bright monolayer of cells enclosing an almost empty lumen (Fig 5). Of note, the organoid culture is polarized, with the apical side oriented toward the inner lumen and the basolateral side in contact with the Matrigel [40]. Organoid expansion is often indistinguishable when comparing non-IBD and IBD cultures (Fig 5). Despite the apparent similar morphology, however, both our study and others demonstrate the existence of intrinsic differences between non-IBD and IBD organoid cultures [12, 41]. These culture-specific signatures are predominantly detectable at a low passage number, since prolonging the culture leads to the progressive loss of intrinsic sample-to-sample differences. This phenomenon was observed, for example, when the expression of inflammatory markers of epithelial origin was analysed in organoid cultures generated from IBD patients versus non-IBD subjects [42, 43]. We thus recommend fixing the number of passages and keeping them low whenever possible when performing comparative studies between cultures. Several alternatives for organoid dissociation have been proposed, including enzymes (e.g., trypsin) and other commercial options (e.g., TrypLE, Gentle Cell Dissociation Reagent). However, we observed that dispase, combined with syringing, is the most gentle method for use on human intestinal organoids. In fact, moderate over-dissociation with this enzyme does not compromise stem cell viability for 3D expansion. Once organoid cultures have been dissociated to single cells, as an alternative to expansion, they can be cryopreserved

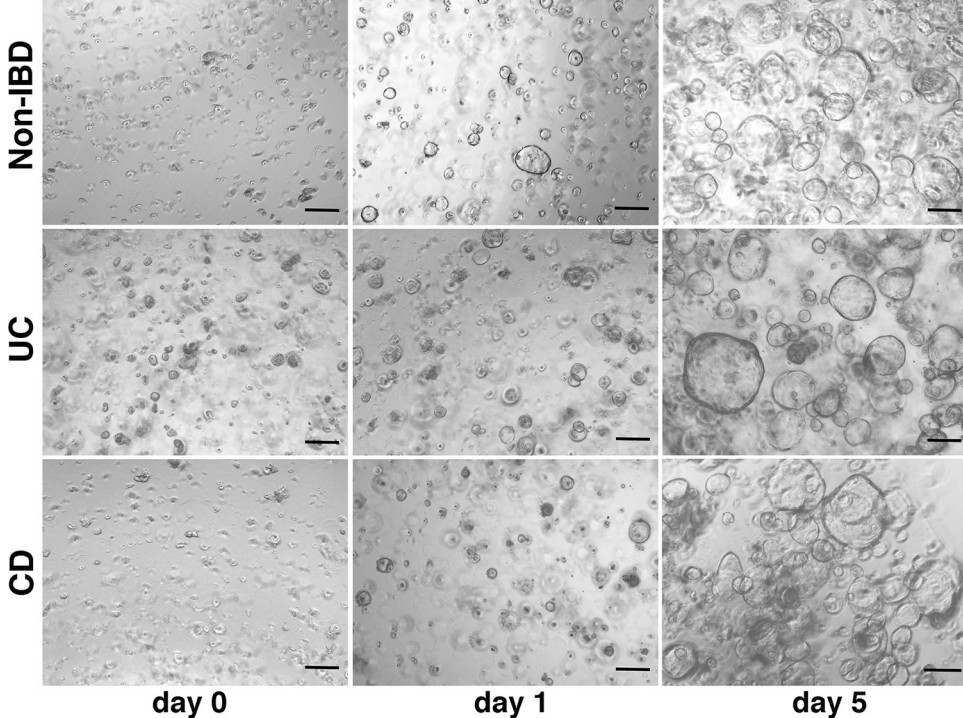

**Fig 5. 3D expansion of colonic ASCs derived from the dissociation of organoid cultures generated from non-IBD and IBD donors.** Once properly dissociated (day 0), organoid cultures can be expanded for 5–6 days before further organoid dilution is required. Samples were derived from the healthy sigmoid colon of a non-IBD donor, the non-inflamed transverse colon of a UC patient, and the not inflamed sigmoid colon of a CD patient. The organoid cultures shown in the figure, all generated from adult subjects, had been previously expanded for two passages. Scale bar: 200 μm.

for long-term storage. Nonetheless, using our protocol we have observed a decrease in the recovery efficiency after thawing, most likely due to a reduction in the content of viable stem cells during freezing and thawing. In support of this finding, previous expression profiling experiments performed in our laboratory show marked transcriptional signature differences between fresh versus previously frozen organoid samples, which included the down-regulation of genes associated with stemness and proliferation, and the up-regulation of genes associated with epithelial differentiation and inflammation in thawed cultures (data not shown). Based on that, we do not recommend combining cohorts of fresh and thawed organoid cultures in projects where comparative analysis between clinical groups of patients are planned. Single cells derived from dissociated organoids are also useful for generating those primary 2D cultures that had already shown high versatility in multiple applications when seeded on conventional plastic plates (S3A Fig), Transwell inserts (S3B Fig) or microfluidic chips [26, 44–47]. In this case, we observed that more aggressive organoid dissociation, like the one mediated by trypsin, is also suitable for growing 2D cultures.

Differentiation of the organoid culture (Fig 1, Step 4 "Organoid differentiation") can be necessary in end-point experiments when specific epithelial cell types, which may not be represented in stem cell-enriched cultures, are to be explored. This process is irreversible and the generated culture must be used within a few days. General differentiation is primarily achieved by removing from the culture those factors that promote the maintenance of the ASC compartment (e.g., WNT3a, RSPO1, SB 202190). In addition, various compounds can be added to the medium to drive the differentiation towards specific intestinal epithelial lineages [48, 49].

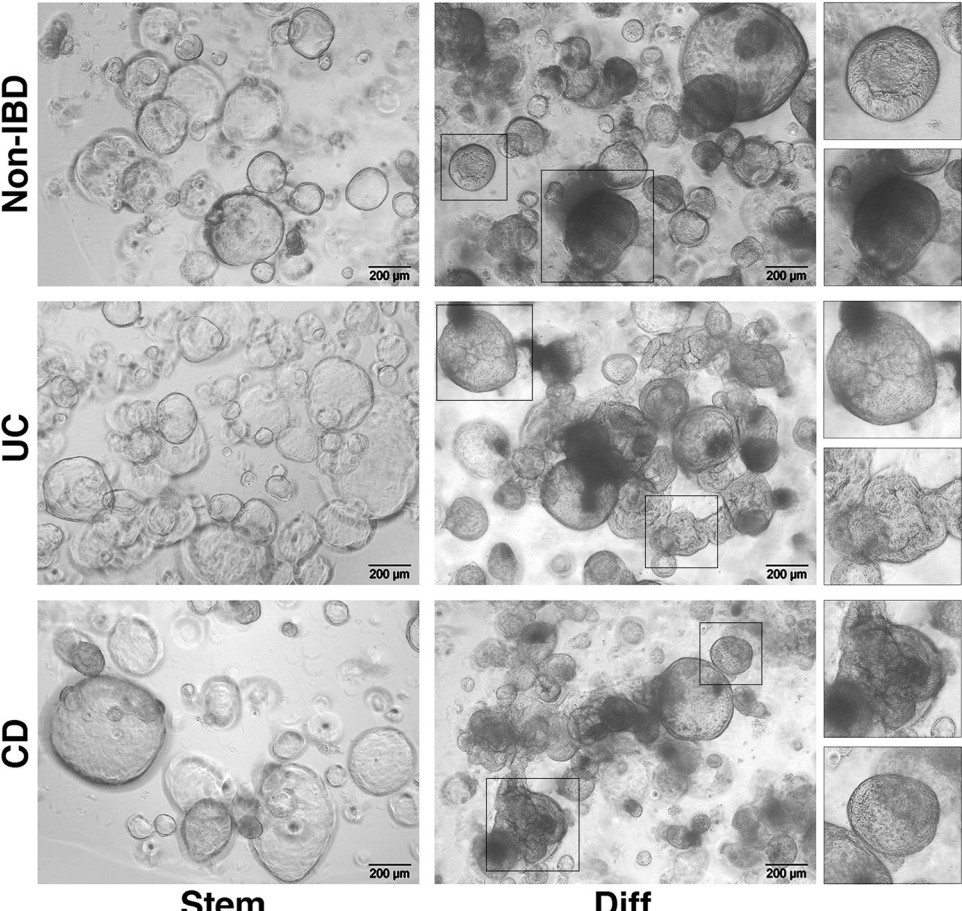

**Fig 6. Differentiation of organoids generated from non-IBD and IBD donors.** The image shows the morphological heterogeneity that can be observed within the differentiated cultures, regardless if there were derived from non-IBD or IBD subjects. "Stem": organoid cultures enriched in the stem cell population on day 5 after passaging; "Diff": organoid cultures on day 5 after differentiation. Samples consist of biopsies obtained from the healthy sigmoid colon of a non-IBD donor, the non-inflamed sigmoid colon of a UC patient, and the mildly inflamed sigmoid colon of a CD patient.

During differentiation, an increasing number of epithelial cells die and are released into the organoid lumen, resulting in the spheres having a darker appearance. In parallel, the walls of both non-IBD and IBD organoids become thicker and their shape more convoluted, when compared with parental stem cell-enriched cultures (Fig 6). The morphological heterogeneity observed within the differentiated organoid cultures, regardless of whether they originated from non-IBD or IBD samples, probably reflects the variability in size and density of the "stem" culture from which they derived.

Given that cell viability significantly decreases during terminal differentiation (usually after day 5), when conducting stimulation experiments we recommend starting the assay 1–2 days prior to complete differentiation.

Finally, we include a troubleshooting guide that lists some of the issues that can arise in each step of the protocol, and possible solutions to these problems (Table 1).

In conclusion, we provide a robust and versatile protocol that can be implemented in any laboratory interested in conducting translational research in IBD using human epithelial organoid technology.

**Table 1. Troubleshooting guide for common problems that may be encountered when generating organoids using this protocol.**

| Step | Issue | Possible reasons | Possible solutions |
|------|-------|------------------|--------------------|
| 1 | Few or no crypts are isolated from the sample | Crypts are isolated from a highly inflamed IBD sample | Avoid isolating crypts from IBD samples with SES-CD [a] > 6 or endoscopic Mayo > 1 |
| | | Mechanical shaking is not sufficient to recover the crypts | If no or few crypts are detectable in the suspension after the first two shaking cycles, incubate the sample in the crypt isolation buffer for an additional 10–15 min at 4°C |
| 2 | Few or no crypts are swelling when cultured | Isolated crypts are not viable | Do not allow excessive mechanical shaking during each shake cycle |
| | | Matrigel lot is not working properly | Change the Matrigel lot. We recommend testing the lot before placing the order of the required number of bottles. |
| | | Organoid growth medium has expired | Discard the old growth medium and prepare fresh medium |
| | | WNT3A conditioned medium is not working | Prepare a new batch of WNT3A conditioned medium |
| 2 | Crypts/organoids are not homogeneously distributed in the Matrigel dome | Crypt/dissociated organoids are not properly resuspended in Matrigel | Pipet the cell/Matrigel mix from time to time to ensure a homogenous suspension (avoid bubbles) |
| 3 | Crypts/organoids do not efficiently expand after passage | Big cell clusters still remain after dissociation | Perform the mechanical dissociation with the syringe until no cell aggregates are visible in the suspension |
| | | Single-cell culture is too diluted | Avoid over-dilution of the dissociated organoids (we never exceed 1:7 dilution). Seed a couple of drops and check the dilution under a microscope before seeding the whole cell/Matrigel sample |
| | | Single-cell culture is too confluent | Avoid under-dilution of the dissociated organoids. Seed a couple of drops and check the dilution at the microscope before seeding the whole cell/Matrigel sample |
| | | Cell viability has been compromised during the dissociation step | Avoid prolonged enzymatic dissociation and syringing during passaging |
| | | Crypts/previously passaged organoids have been kept in culture for too long | Do not leave the crypt culture longer than 3 days nor the organoid culture for longer than 6–7 days |
| | | Matrigel lot is not working properly | Change the Matrigel lot. We recommend testing the lot before placing an order for the required number of bottles |
| | | Organoid growth medium has expired | Discard the old growth medium and prepare fresh medium |
| | | WNT3A conditioned medium is not working | Prepare a new batch of WNT3A conditioned medium |
| | | Nutrient exhaustion | Refresh culture medium every 2–3 days. |
| 2–3 | Matrigel dome does not form/is unstable over time | Dispase has not been completely removed | Perform the number of washes suggested in the protocol |
| | | Plate has not been warmed | Pre-warm the plate in the incubator for at least 2 hours prior to use |
| | | Matrigel lot is not working properly | Change the Matrigel lot. We recommend testing the lot before placing an order for the required number of bottles |
| | | The plastic of the culture plate is not compatible | We recommend using plates from Corning (e.g., Cod. 3548 for a 48-well plate) |

[a] SES-CD: Simple Endoscopic Score for Crohn's Disease

In both examples, organoids were generated from surgical samples derived from two non-IBD donors and dissociation was performed by dispase treatment according to the described protocol. Scale bar: 200 μm.

## Supporting information

**S1 File. Step-by-step protocol.** Also available on protocols.io.
(PDF)

**S1 Fig. Bright-field image of a representative human colonic crypt.** In this image, the bottom of the crypt containing the adult stem cell (ASC) niche and the apical crypt, where

terminally differentiated cells are located, have been highlighted. This picture was taken from a supernatant enriched in crypts isolated from biopsies of the descending colon of a non-IBD donor. Scale bar: 100 μm.
(TIF)

**S2 Fig. Accumulation of dead cells in a colonic crypt culture.** The viability of a crypt culture rapidly decreases over time due to the release of dead cells and debris into the lumen. This sample was obtained from the mildly inflamed sigmoid colon of an adult CD patient. Scale bar: 100 μm.
(TIF)

**S3 Fig. Examples of 2D cultures derived from the dissociation of 3D organoids.** After organoid dissociation, epithelial single cells/small clusters of cells were counted using conventional methods and seeded on different supports. To promote cell adhesion and growth, a coating layer of diluted Matrigel was added. A) 2D epithelial culture at low confluence after seeding $5x10^4$ cells on a 48-well plate (pre-coated with 1:20 Matrigel). Image was taken 24h after cell seeding; B) Fully confluent 2D epithelial culture after seeding $5x10^4$ cells on a 0.33 cm$^2$ Transwell insert (0.4 μm pore, pre-coated with 1:40 Matrigel). Image was taken 7 days after cell seeding.
(TIF)

## Acknowledgments

We thank Eduard Batlle and Peter Jung for the scientific support during the establishment of the proposed protocol in our laboratory. We thank the IBD Unit, Endoscopy and Pathology Departments at Hospital Clinic Barcelona, Hospital Universitari Mútua de Terrassa and Hospital Sant Joan de Déu Barcelona for providing us with the samples used in accordance with the proposed protocol, and all the patients for their selfless participation. We also thank Marta Martinez for technical support, and Joe Moore for editorial assistance.

## Author Contributions

**Conceptualization:** Isabella Dotti.

**Data curation:** Isabella Dotti, Aida Mayorgas.

**Formal analysis:** Isabella Dotti, Aida Mayorgas.

**Funding acquisition:** Azucena Salas.

**Investigation:** Isabella Dotti, Aida Mayorgas.

**Methodology:** Isabella Dotti, Aida Mayorgas.

**Resources:** Azucena Salas.

**Supervision:** Azucena Salas.

**Writing – original draft:** Isabella Dotti.

**Writing – review & editing:** Azucena Salas.

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
