## [Decision Letter · Decision Letter 0]

18 Jul 2022

PONE-D-22-12935

Generation of human colon organoids from healthy and inflammatory bowel disease mucosa

PLOS ONE

Dear Dr. Dotti,

Thank you for submitting your manuscript to PLOS ONE. After careful consideration, we feel that it has merit but does not fully meet PLOS ONE’s publication criteria as it currently stands. Therefore, we invite you to submit a revised version of the manuscript that addresses the points raised during the review process.

We look forward to receiving your revised manuscript.

Kind regards,

Xiaonan Han, Ph.D.

Academic Editor

PLOS ONE

Journal Requirements:

2. We note that your study involved tissue/organ donations. Please provide the following information regarding tissue/organ donors analyzed in your study.

1. Please provide the source(s) of the donors, including the institution name and a non-identifying description of the donor(s).

2. Please state in your response letter and ethics statement whether the donors for this study involved any vulnerable populations; for example, tissue/organs from prisoners, subjects with reduced mental capacity due to illness or age, or minors.

- If a vulnerable population was used, please describe the population, justify the decision to use tissue/organ donations from this group, and clearly describe what measures were taken in the informed consent procedure to assure protection of the vulnerable group and avoid coercion. 

- If a vulnerable population was not used, please state in your ethics statement, “None of the donors was from a vulnerable population and all donors or next of kin provided written informed consent that was freely given.”

3. In the Methods, please provide detailed information about the procedure by which informed consent was obtained from organ/tissue donors or their next of kin. In addition, please provide a blank example of the form used to obtain consent from donors, and an English translation if the original is in a different language.

4. Please indicate whether the donors were previously registered as organ donors. If tissues/organs were obtained from deceased donors or cadavers, please provide details as to the donors’ cause(s) of death.

5. Please provide the participant recruitment dates and the period during which transplant procedures were done (as month and year). Please also upload a) the original (signed or stamped) ethics approval document and b) the original study protocol as ‘Other’ files.

6. Please discuss whether medical costs were covered or other cash payments were provided to the family of the donor. If so, please specify the value of this support (in local currency and equivalent to U.S. dollars).

Reviewers' comments:

Reviewer's Responses to Questions

**Comments to the Author**

1. Does the manuscript report a protocol which is of utility to the research community and adds value to the published literature?

Reviewer #1: Yes

Reviewer #2: Yes

2. Has the protocol been described in sufficient detail?

Descriptions of methods and reagents contained in the step-by-step protocol should be reported in sufficient detail for another researcher to reproduce all experiments and analyses. The protocol should describe the appropriate controls, sample sizes and replication needed to ensure that the data are robust and reproducible.

Reviewer #1: Yes

Reviewer #2: Yes

3. Does the protocol describe a validated method?

Reviewer #1: Yes

Reviewer #2: Yes

4. If the manuscript contains new data, have the authors made this data fully available?

Reviewer #1: Yes

Reviewer #2: Yes

**5. Is the article presented in an intelligible fashion and written in standard English?**

Reviewer #1: Yes

Reviewer #2: Yes

6. Review Comments to the Author

Reviewer #1: General comments: The manuscript “Generation of human colon organoids from healthy and inflammatory bowel disease Mucosa” detailed summarized a protocol of generation 3D organoids from patient’s biopsies including troubleshooting part and the tips of increasing the success rate of generating stable organoids. By using this protocol, these organoids can be maintained for at least 6 passages without the decrease in the expansion rate. Here are some suggestions and comments to further improve the manuscript.

Comment 1: Author mentioned that these patient-derived 3D organoids can be used for generating 2D structures. Are there any figures showing the 2D structures established from 3D organoids?

Comment 2: Please briefly explain the percentage in the pie chart (Figure 2). Which percentages refer to the successful generation of 3D organoids(yes) and which ones refer to the failure(no)? Readers might be confused with it.

Comment 3: Lack the cryopreservation part. Can these 3D organoids be frozen and thawed? Whether thawed organoids showed stable expansion?

Comment 4: Figure 6 showed the morphology of the 3D organoid during differentiation in 3 groups. Adding higher magnification views exhibiting the budding structure of the organoid would be better. The low magnification in figure 6 only showed a clear view of the lumen.

Reviewer #2: On the Lab Protocol manuscript number; PONE-D-22-12935 entitled “Generation of human colon organoids from healthy and inflammatory bowel disease mucosa” the paper generates the following kinds of data;

1.Compared the organoids' growth and stability Ulcerative colitis and Crohn’s Disease are chronic inflammatory bowel diseases (IBD).

2. Described in detail a reproducible protocol to generate Matrigel-embedded epithelial organoids from ASCs of non-IBD and IBD donors using small colonic biopsies, including steps for its optimization.

3. Surgical biopsy samples were used.

4.Epithelial organoids can be expanded over several passages, thereby generating a large quantity of viable cells that can be used in multiple downstream analyses including genetic, transcriptional, proteomic and/or functional studies. In addition, 3D cultures generated using our protocol are suitable for the establishment of 2D cultures, which can model relevant cell-to-cell interactions that occur in IBD mucosa.

This paper is clearly written and well organized. the introduction and background are reasonable given the premise of the paper. figures and tables are comprehensive and helpful.

Minor issues need to be dressed

# The author recommend using at least 5-6 donors when in vitro stimulation experiments are planned, and at least 12-15 donors per group when comparative analyses between cohorts of patients. In addition, as organoids derived from different segments of the intestinal tract show intrinsic differences (11, 33), also recommend, whenever possible, using biopsies from the same anatomical region to minimize this variability.

To modify the clinical applications of this method such as using internal control by collecting the samples from the patient’s colon healthy areas.

# How many times successfully step 3 Step 3 can be repeated without changes in molecular features?

7. PLOS authors have the option to publish the peer review history of their article (what does this mean?). If published, this will include your full peer review and any attached files.

Reviewer #1: No

Reviewer #2: No

---

## [Author Response · Author response to Decision Letter 0]

5 Sep 2022

"Response to Reviewers" has been also uploaded as an attachment. 

'Response to Reviewers'

We thank the academic editor and the two reviewers for their comments on our manuscript entitled “Generation of human colon organoids from healthy and inflammatory bowel disease mucosa”.

Provided below is a point-by-point reply to the questions raised by the editor and the reviewers.

Editor’s comments:

R1: We thoroughly reviewed the manuscript to ensure compliance with the journal requirements.

2. We note that your study involved tissue/organ donations. Please provide the following information regarding tissue/organ donors analyzed in your study.

1. Please provide the source(s) of the donors, including the institution name and a non-identifying description of the donor(s).

R1: This information has been added in the “Materials and Methods” section.

2. Please state in your response letter and ethics statement whether the donors for this study involved any vulnerable populations; for example, tissue/organs from prisoners, subjects with reduced mental capacity due to illness or age, or minors. If a vulnerable population was used, please describe the population, justify the decision to use tissue/organ donations from this group, and clearly describe what measures were taken in the informed consent procedure to assure protection of the vulnerable group and avoid coercion.

R2: In the present study we describe a method for generating organoids from both adult and pediatric donors. For the latter, we used a database of organoids established from 45 intestinal samples derived from a total of 45 pediatric subjects (12 non-IBD controls; 32 CD patients; 1 UC patient). The age of pediatric subjects ranged between 8 and 16 years. Of these, we further reduced the collection of intestinal biopsies to 4, which in our hands is the minimum number required for growing viable ex vivo organoid cultures according to the protocol described in our manuscript.

Our interest in exploring the role of the intestinal epithelial barrier in pediatric IBD patients stems from the fact that while early-onset IBD represents a distinct disease in terms of phenotype, location and behaviour, not much is known about the role of the epithelium in these patients. In addition, we benefitted from an established collaboration with the pediatric hospital Sant Joan de Deu in Barcelona and the IBD Unit at Hospital Clinic. As a result of this partnership, in 2019 our group was awarded an ECCO Grant for studying the interplay between the epithelial barrier and the mucosal environment in pediatric CD patients compared to a cohort of adult subjects. 

The population of pediatric donors has been described in detail in the “Materials and Methods” section. In addition, in the “Ethics declarations” statement we specified how the informed consent was obtained from these donors.

3. In the Methods, please provide detailed information about the procedure by which informed consent was obtained from organ/tissue donors or their next of kin. In addition, please provide a blank example of the form used to obtain consent from donors, and an English translation if the original is in a different language.

R3: information on how informed consent was obtained from both adult and pediatric donors has been added in the “Materials and Methods” section.

Two blank informed consents have been uploaded: one is common to Hospital Clinic Barcelona and Hospital Mutua de Terrassa (File “1. Informed consent Adult”), while the other is specific for Hospital Sant Joan de Deu (File “2. Informed consent Pediatric”). File “3. Approval” shows the approval for sharing the same consent between Hospital Clinic and Mutua de Terrassa. Moreover, for this consent the last version has been uploaded since the original version was revised in accordance with Spanish legislation for data protection active at the time. If required, we have the previous versions available. For each consent, an English translation has also been provided (File “4. Informed consent Adult ENG” and File “5. Informed consent Pediatric ENG”). 

4. Please indicate whether the donors were previously registered as organ donors. If tissues/organs were obtained from deceased donors or cadavers, please provide details as to the donors’ cause(s) of death.

R4: We added this information to the “Materials and Methods” section.

5. Please provide the participant recruitment dates and the period during which transplant procedures were done (as month and year). Please also upload a) the original (signed or stamped) ethics approval document and b) the original study protocol as ‘Other’ files.

R5: We added the recruitment dates to the “Materials and Methods” section.

The presented lab protocol is based on a database of samples collected over the last 5 years (between 2016 and 2021). We uploaded the original ethics approval document for each study (File “6. Ethics approval 1”; File “7. Ethics approval 1_amendment”; File “8. Ethics approval 2”; File “9. Ethics approval 3_HCB”; File “10. Ethics approval 3_SJD”). The same has been done for the original study protocols (File “11. Study protocol 1”; File “12. Study protocol 2”; File “13. Study protocol 3”).

6. Please discuss whether medical costs were covered or other cash payments were provided to the family of the donor. If so, please specify the value of this support (in local currency and equivalent to U.S. dollars).

R6: We added this information to the “Materials and Methods” section.

R3: Based on the reviewers’ comments, we updated the protocol uploaded to protocol.io. The new DOI and link for the reviewers have been included in the revised manuscript in the “Associated content” and “Materials and Methods” sections. We will share the protocol in protocol.io once our manuscript is accepted for publication.

R4: All this information has been added both to the “Materials and Methods” section and to the “Ethics declarations” statement. We also indicated that specific informed consents were provided for both pediatric and adult donors.

R5: We checked thoroughly and included 2 new references in the manuscript in response to the comments of Reviewer #2 in order to support our reply. The references and their associated numbers in the revised manuscript are the following: 

34. Olafsson S, McIntyre RE, Coorens T, Butler T, Jung H, Robinson PS, et al. Somatic Evolution in Non-neoplastic IBD-Affected Colon. Cell. 2020;182(3):672-84 e11.

43. Porpora M, Conte M, Lania G, Bellomo C, Rapacciuolo L, Chirdo FG, et al. Inflammation Is Present, Persistent and More Sensitive to Proinflammatory Triggers in Celiac Disease Enterocytes. Int J Mol Sci. 2022;23(4).

We have not cited papers that have been retracted.

Reviewers’ comments:

Reviewer # 1

General comments: The manuscript “Generation of human colon organoids from healthy and inflammatory bowel disease Mucosa” detailed summarized a protocol of generation 3D organoids from patient’s biopsies including troubleshooting part and the tips of increasing the success rate of generating stable organoids. By using this protocol, these organoids can be maintained for at least 6 passages without the decrease in the expansion rate. Here are some suggestions and comments to further improve the manuscript.

We would like to thank the reviewer for the time devoted to evaluate our manuscript and the constructive criticism provided.

1: Author mentioned that these patient-derived 3D organoids can be used for generating 2D structures. Are there any figures showing the 2D structures established from 3D organoids?

R1: As suggested, we included a Supplementary Figure (S3 Fig) and the corresponding legend showing two examples of 2D culture generated from 3D organoids in our laboratory, the first one seeding epithelial single cells on a conventional 48-well plate, and the other seeding them on a Transwell. The main text has been modified on page 14, line 320, of the revised manuscript.

2: Please briefly explain the percentage in the pie chart (Figure 2). Which percentages refer to the successful generation of 3D organoids (yes) and which ones refer to the failure (no)? Readers might be confused with it.

R2: We completely agree with the reviewer’s observation, and have modified Figure 2. We slightly modified the legend of the figure by substituting “yes” and “no” with more appropriate terms to make it more understandable to the reader. The new terms (“Expansion” and “No expansion”) have also been included in the pie chart legend. 

3: Lack the cryopreservation part. Can these 3D organoids be frozen and thawed? Whether thawed organoids showed stable expansion?

R3: As suggested by the reviewer, we included a comment about cryopreservation in our manuscript. In addition, we generated a new version of the protocol with the updated DOI (DOI: dx.doi.org/10.17504/protocols.io.rm7vz3wr4gx1/v2;

-Private link for reviewers:

https://www.protocols.io/private/F13BA2CE694A8C5FFCF0EBD83DC6FBE1

to be removed before publication-) that includes steps for freezing and thawing (new step 51.1). The new links have been introduced in “Associated content” and “Materials and methods” sections.

Of note, although frozen organoid cultures can be thawed and expanded using our protocol, the efficiency of culture expansion after thawing is lower than that observed when growing fresh cultures. Moreover, we noticed marked expression signature differences between thawed and fresh organoid samples, as indicated by the following Principal Component Analysis of a microarray experiment performed on frozen/thawed (n=3) versus fresh cultures (n=6): 

FIGURE

Among the top deregulated genes, frozen cultures showed a decrease in LGR5, ASCL2 and CDCA7 (markers of stemness and proliferation), together with the up-regulation of TFF2 and SLC26A3 (markers of differentiation), and DUOX2 (marker of inflammation): 

FIGURE

These observations suggest that the cryopreservation procedure can affect epithelial stem cell viability (and thus the efficient recovery) of the thawed culture. 

For these reasons, we do not recommend mixing fresh and thawed organoids in projects involving, for example, comparative analysis between clinical groups of patients. These observations have been included in the revised manuscript on page 13, lines 306-217.

4: Figure 6 showed the morphology of the 3D organoid during differentiation in 3 groups. Adding higher magnification views exhibiting the budding structure of the organoid would be better. The low magnification in figure 6 only showed a clear view of the lumen.

R4: As suggested by the reviewer, we modified Figure 6 and included magnified inserts showing the intricate 3D structures of the differentiated organoids in comparison with the “stem” cultures. In particular, we included inserts showing organoids with different shapes and sizes within each culture, thereby focusing on the typical morphological heterogeneity which characterizes the differentiated cultures, regardless of whether they were generated from non-IBD or IBD samples. The main text (Page 14, lines 332-337) and the legend of Figure 6 have been modified accordingly.

Reviewer # 2

On the Lab Protocol manuscript number; PONE-D-22-12935 entitled “Generation of human colon organoids from healthy and inflammatory bowel disease mucosa” the paper generates the following kinds of data;

1.Compared the organoids' growth and stability Ulcerative colitis and Crohn’s Disease are chronic inflammatory bowel diseases (IBD).

2. Described in detail a reproducible protocol to generate Matrigel-embedded epithelial organoids from ASCs of non-IBD and IBD donors using small colonic biopsies, including steps for its optimization.

3. Surgical biopsy samples were used.

4. Epithelial organoids can be expanded over several passages, thereby generating a large quantity of viable cells that can be used in multiple downstream analyses including genetic, transcriptional, proteomic and/or functional studies. In addition, 3D cultures generated using our protocol are suitable for the establishment of 2D cultures, which can model relevant cell-to-cell interactions that occur in IBD mucosa.

This paper is clearly written and well organized. The introduction and background are reasonable given the premise of the paper. figures and tables are comprehensive and helpful.

Minor issues need to be dressed

We would like to thank the reviewer for the positive feedback and suggestions provided.

1: The author recommend using at least 5-6 donors when in vitro stimulation experiments are planned, and at least 12-15 donors per group when comparative analyses between cohorts of patients. In addition, as organoids derived from different segments of the intestinal tract show intrinsic differences (11, 33), also recommend, whenever possible, using biopsies from the same anatomical region to minimize this variability.

To modify the clinical applications of this method such as using internal control by collecting the samples from the patient’s colon healthy areas.

R1: We thank the reviewer for her/his suggestion. A recent work showed that the average mutation rate of colonic IBD epithelial cells affected by prolonged inflammation is higher than that observed in healthy epithelium; several of these mutations involve IL-17 and Toll-like receptor pathways (Olafsson S et al, Cell. 2020 Aug 6;182(3):672-684.e11). Similarly, another study showed that organoid cultures derived from affected IBD epithelium accumulate somatic mutations more frequently than organoids from non-IBD mucosa (Nanki, K et al. Nature 577, 254–259 (2020)). Again, several of the accumulated mutations are associated with the IL-17 signaling pathway. Interestingly, in this work the authors stress how the accumulation of mutations is dependent on the presence of chronic inflammation rather than disease location. Thus, the use of organoids from the affected and unaffected areas of a patient with a long history of disease can be helpful for exploring the impact of the inflammatory microenvironment on the behaviour of the IBD epithelium.

We added this information on page 6, lines 129-137, of the revised manuscript and the new citation by Olafsson S et al. has been included in the text with number 34.

2: How many times successfully Step 3 can be repeated without changes in molecular features?

R2: The reviewer raises a relevant point. Indeed, long-term in vitro expansion unavoidably introduces progressive expression changes in the analysed culture. We have direct experience with this phenomenon. Indeed, by analysing the expression levels of a panel of typical intestinal epithelial markers in cultures expanded for as little as 3-4 passages, we could observe progressive transcriptional alterations of these markers over passages. In line with our observations, other studies (Arnauts K et al, Gastroenterology. 2020, 159(4):1564-7; Porpora M et al, Int J Mol Sci. 2022 Feb 10;23(4):1973) have shown that prolonged culture leads to the loss of intrinsic features, like those associated with the response of IBD epithelium to the pro-inflammatory mucosal environment. 

Based on these findings, we propose that a way to partially reduce the bias introduced by the organoid passaging on the molecular signatures is to maintain a low and consistent number of passages throughout the study when performing comparative analysis between cultures. This concept has been applied in our paper (Dotti I et al, Gut, 2017), in which we compared organoids derived from non-IBD and IBD mucosa using cultures at early expansion steps (all samples had only been expanded once in that particular study).

This has been mentioned on page 13, lines 296-301, of the revised manuscript and the work published by Porpora et al has been included in the text with number 43.

---

## [Editor Report · Decision Letter 1]

2 Oct 2022

Generation of human colon organoids from healthy and inflammatory bowel disease mucosa

PONE-D-22-12935R1

Dear Dr. Salas,

We’re pleased to inform you that your manuscript has been judged scientifically suitable for publication and will be formally accepted for publication once it meets all outstanding technical requirements.

Kind regards,

Xiaonan Han, Ph.D.

Academic Editor

PLOS ONE

Additional Editor Comments (optional):

Dr. Salas,

After reading your response to comments, I am very satisfied with them. Please finalize your manuscript based upon PLOSONE instruction.

Best regards

Dr. Xiaonan Han
---

## [Editor Report · Acceptance letter]

19 Oct 2022

PONE-D-22-12935R1 

Generation of human colon organoids from healthy and inflammatory bowel disease mucosa 

Dear Dr. Salas:

I'm pleased to inform you that your manuscript has been deemed suitable for publication in PLOS ONE. Congratulations! Your manuscript is now with our production department. 

Kind regards, 

on behalf of

Dr. Xiaonan Han 

Academic Editor

PLOS ONE